# Cache-Based Privacy Preserving Solution for Location and Content Protection in Location-Based Services

**DOI:** 10.3390/s20164651

**Published:** 2020-08-18

**Authors:** Yuanbo Cui, Fei Gao, Wenmin Li, Yijie Shi, Hua Zhang, Qiaoyan Wen, Emmanouil Panaousis

**Affiliations:** 1State Key Laboratory of Networking and Switching Technology, Beijing University of Posts and Telecommunications, Beijing 100876, China; cuiyb@bupt.edu.cn (Y.C.); gaof@bupt.edu.cn (F.G.); yijieshi2000@bupt.edu.cn (Y.S.); zhanghua_288@bupt.edu.cn (H.Z.); wqy@bupt.edu.cn (Q.W.); 2School of Computing and Mathematical Sciences, University of Greenwich, London SE10 9LS, UK; e.panaousis@greenwich.ac.uk

**Keywords:** cache, location-based services, location privacy, query privacy, mobile peer-to-peer network

## Abstract

Location-Based Services (LBSs) are playing an increasingly important role in people’s daily activities nowadays. While enjoying the convenience provided by LBSs, users may lose privacy since they report their personal information to the untrusted LBS server. Although many approaches have been proposed to preserve users’ privacy, most of them just focus on the user’s location privacy, but do not consider the query privacy. Moreover, many existing approaches rely heavily on a trusted third-party (TTP) server, which may suffer from a single point of failure. To solve the problems above, in this paper we propose a Cache-Based Privacy-Preserving (CBPP) solution for users in LBSs. Different from the previous approaches, the proposed CBPP solution protects location privacy and query privacy simultaneously, while avoiding the problem of TTP server by having users collaborating with each other in a mobile peer-to-peer (P2P) environment. In the CBPP solution, each user keeps a buffer in his mobile device (e.g., smartphone) to record service data and acts as a micro TTP server. When a user needs LBSs, he sends a query to his neighbors first to seek for an answer. The user only contacts the LBS server when he cannot obtain the required service data from his neighbors. In this way, the user reduces the number of queries sent to the LBS server. We argue that the fewer queries are submitted to the LBS server, the less the user’s privacy is exposed. To users who have to send live queries to the LBS server, we employ the l-diversity, a powerful privacy protection definition that can guarantee the user’s privacy against attackers using background knowledge, to further protect their privacy. Evaluation results show that the proposed CBPP solution can effectively protect users’ location and query privacy with a lower communication cost and better quality of service.

## 1. Introduction

With the rapid advances of mobile devices and wireless communication, Location-Based Services (LBSs) have been a vital part of people’s daily life in recent years. A growing number of people are downloading location-based applications from the App Store or Google Play Store through their mobile devices (e.g., smartphones). With such applications, users can easily issue queries to the LBS servers and obtain the corresponding service data. For example, users can look for the metro stations or the price information of hotels nearby [1]. Thus, LBSs have greatly changed the way people live.

Although LBSs provide a wide variety of conveniences to users, they also pose a serious threat to users’ privacy and security. Normally, when a user wants to obtain a service, he needs to submit his exact location and queried interest to an untrusted LBS server. As a result, the server has all the information about the user, and may track the user directly or release his personal information to others  [2]. Such information, however, is extremely sensitive, and may endanger the user’s physical security if they fall in the wrong hands. Thus, we need to pay more attention to protecting the user’s privacy.

To address the privacy issue in LBSs, many approaches have been proposed over the past few years [1,2,3,4,5,6,7,8,9,10,11,12,13,14,15,16,17,18,19,20,21,22,23,24,25,26,27,28,29,30,31,32,33,34]. Generally, these approaches can be roughly classified as the:(a)*False locations*. Users protect their privacy by sending either fake locations [18] or their real locations with a set of fake locations, called dummies [1,12,19,20] to the LBSs server.(b)*Spatial cloaking*. The basic idea of the spatial cloaking is to blur a user’s exact location into a cloaked area that satisfies the user’s privacy requirements [11,13,14,17].(c)*Space transformation*. A user utilizes a space-filling curve (e.g., Hilbert Curve) to transform his exact location into another space to protect the user’s privacy [25,26].

Actually, these approaches can always provide privacy preservation to users by using a well-known privacy metric (e.g., k-anonymity [11,12,13,14,15,19] or entropy [1,2,20,21,23]). Moreover, since the emergence of blockchain, some works, such as [27,28,29,30,31,32], combine the LBS and blockchain technology together to protect users’ location privacy. The combination is now a new trend and has received more and more attention recently.

Existing works, however, also have some limitations. First, most of them just focus on the user’s location privacy [1,11,12,13,14,15,16,17,19,20,21,22,23]. Actually, user privacy in LBSs includes two aspects, i.e., *location privacy* and *query privacy*, and they are closely related. There is the possibility that compromising one of them may lead to the failure of the other [26]. Therefore, we need to protect location privacy and query privacy simultaneously. Secondly, many previous studies [11,13,14,15,16,26] introduce a trusted third party (TTP) server, called the *Location Anonymizer*, to preserve the user’s privacy in an LBS environment. However, the TTP server possesses the following shortcomings: (1) The TTP server could suffer from a single point of failure. If the adversary gains control of it, the privacy of all users will be compromised; (2) The TTP server may be the performance bottleneck of the system, since all the submitted queries have to go through the TTP server [20]; (3) In practice, it is impossible to find a third party that can be fully trusted by all users. Thirdly, many users may query for the same interest in reality, and the LBS server thus has to answer the same service data to the users repeatedly, which greatly increases the exposing risk of users’ privacy.

To solve the aforementioned problems, in this paper we propose a Cache-Based Privacy-Preserving (CBPP) solution for LBSs. Different from the previous approaches, our proposed CBPP solution preserves the user’s location privacy and query privacy simultaneously, and avoids the problem of TTP server by having users collaborate with each other in a mobile peer-to-peer (P2P) environment. The key idea of the proposed CBPP solution is that each user caches the service data obtained from the LBS server or the neighbors in his mobile device, and uses the cached data to answer the later queries issued by others. Specifically, when a user wants LBSs, he needs to broadcast a query to his neighbors first to seek for an answer. The user only contacts the LBS server if he cannot get the required service information from their neighbors. In this way, a user reduces the number of queries sent to the LBS server, and remains hidden from the LBS server. Our main contributions in this paper can be summarized as follows:We provide the Cache-Based Privacy-Preserving (CBPP) solution for users in LBSs, which can protect the user’s location privacy and query privacy simultaneously. Meanwhile, we avoid the problem of TTP server by having users collaborate with each other in a mobile peer-to-peer (P2P) environment.We reduce the number of queries sent to the untrusted LBS by using the caching, which not only protects the user’s privacy effectively against the LBSs server, but also improves the user’s query efficiency and cache hit ratio.We analyze the availability and the security of proposed CBPP solution, which show that the proposed CBPP solution is a much more practical way to protect users’ privacy in LBSs.

The rest of the paper is organized as follows. We discuss related work in Section 2. Section 3 presents some preliminaries of this paper. We present our CBPP scheme in Section 4. The security analysis and the evaluation results are shown in Section 5 and Section 6, respectively. We describe the conclusion and future work in Section 7.

## 2. Related Work

### 2.1. Present Research Situation

Privacy in LBSs has been one of the most popular research topics recently, and more and more privacy-preserving solutions have been proposed to ensure the user’s safety. Actually, most solutions provide anonymity on users’ exact locations to hide their location information. The objective of anonymity is to separate a user’s identity from his location by using various techniques. Among various anonymizing techniques, *k-anonymity*, which was first introduced into LBSs by Gruteser et al. in [11], is the most widely used metric. To achieve *k-anonymity*, a trusted middleware (i.e., *Location Anonymizer*) was used between the user and the LBS server in [11]. When a query is submitted to the LBS server, the location anonymizer enlarges the queried location into a cloaking area, which contains the user and at least k−1 others. Then the location anonymizer reports this area as the user’s location to the LBS server. Since any entity inside the cloaking area could be the user, the LBS server cannot distinguish the real user from others. However, this work assumes a static *k* value, which cannot satisfy the users with different privacy requirement (i.e., *k* value). To address this problem, Gedik et al. considered a personalized *k-anonymity* and proposed Clique-Cloak scheme in [14], which can make users adjust their anonymity degree. Unfortunately, all approaches above require the user to expose his exact location to the *location anonymizer*, which may suffer from the single point of failure on either system performance or user privacy [2].

To avoid the *location anonymizer*, Kido et al. [12] proposed the dummy locations-based solution to achieve *k-anonymity* in LBSs. In this work, a user protects his privacy by reporting his exact location with a set of fake locations, termed dummies, directly to the LBS server to achieve the *k-anonymity*. However, they use a random walk model to generate dummy locations, which cannot ensure the designed privacy level when the adversary (e.g., the LBS server) has some background knowledge. For example, some randomly generated dummy locations may fall at some unlikely locations such as lakes, oceans, swamps, and rugged mountains, and can be easily filtered out by the adversary. The desired *k-anonymity* thus fails. Later, Lu et al. [19] proposed two dummy locations generating algorithms, i.e., *CirDummy* and *GridDummy*, which also provide *k-anonymity* to users by considering the privacy area. However, this work still cannot really guarantee the desired degree of anonymity when the adversary has background knowledge. Then, Niu et al. [1,20,21,23] proposed a set of solutions to address this problem. Actually, all of these solutions choose dummy locations carefully based on the entropy metric, which significantly improves the privacy level against the adversary who has background knowledge. Nevertheless, all the above mentioned approaches pay much attention to the user’s location privacy, but neglect the query privacy.

### 2.2. Most Related Work to Our Solution

The most related works to ours are shown in [21,22]. Both of the works are based on the caching technique, and the location privacy is maintained by the user-collaborative approach. The basic idea of the cache-based schemes is that users only contact the LBS server if they cannot get the required service data from their neighbors, who cache the used service data in their buffers. However, there are some drawbacks in these works: (1) Both of them do not preserve the user’s query privacy, which is also pretty important in LBSs; (2) In [22], it is hard to protect the first user’s privacy for each query, because the user has to send a live query to the untrusted LBS server to obtain the required service. Although [21] solves this problem, it still has lower cache hit ratio.

## 3. Preliminaries

In this section, we first introduce the motivation, and then present the basic idea of our solution.

### 3.1. Motivation

In an LBS environment, when a user wants to enjoy services, he has to send a location-related query to the untrusted LBS server, and then the latter returns the corresponding service data to the user based on the location information and query interest of the user. However, there exist some drawbacks with this pattern. First, due to the fact that each query sent to the LBS server contains location information and query interest of the user, the LBS server could know where the user is and what kind of query the user submits, which may threaten user’s privacy and security. Even worse, the LBS server can infer some private information of the user based on the query, such as his identity, home location, lifestyle habit, even health condition, etc., [26]. Secondly, in the real world, some users may query for the same interest, which leads to the LBS server has to reply them with the same answer repeatly. For example, there may be many people at the same metro station to submit queries with the same contents to the LBS server (e.g., how to arrive at some metro station or shopping mall). As a result, the LBS server has to respond to the same service data repeatedly, which obviously leads to low efficiency.

To solve the first problem, a straightforward method is to use *k-anonymity* which we mentioned above, which renders the LBS server unable to distinguish the real user from the other k−1 users. However, on the one hand, it is not always easy to find k−1 users at any time and any place to achieve *k-anonymity*, and on the other hand, most existing studies just protect the user’s location privacy by using *k-anonymity*, but do not consider the user’s query privacy. Actually, we can protect the user’s query privacy by having the query content achieving *k-anonymity*. In other words, we can send the real query with the other k−1 queries to the LBS server. However, due to the background knowledge the LBS server has, *k-anonymity* does not guarantee the user’s query privacy. Fortunately, *TTcloak* [2] shows us an effective method, i.e., *l-diversity* [35], to protect the user’s query privacy. We employ the *l-diversity* to protect the user’s location privacy and query privacy simultaneously. To solve the second problem, cache technique [22] is introduced into the LBSs. The reason for the adoption of cache includes the following two aspects, (1) some users may query for the same interest in reality; (2) the LBS related service data generally has longer lifetime [36]. Thus, users can cache the previous service data in their mobile device to answer future queries. Moreover, cache techniques also reduce the number of queries sent to the LBS server, which further enhances the user’s privacy.

### 3.2. Our Basic Idea

In this paper, we integrate the *l-diversity* and cache techniques to solve the aforementioned problems. Actually, our work protects the user’s privacy in LBSs by relying on the collaboration between users in mobile peer-to-peer (P2P) environments. Through such a user-cooperative approach, our solution not only improves users’ privacy, but also avoids the problems of TTP server. Specifically, when a user needs LBSs, he sends a query to his neighbors first. Since each user keeps a buffer in his mobile device to record the service data, neighbors search their buffers to seek for an answer after they receive the query.

(1) If the neighbors cache the corresponding service data, then the neighbors pass the data back to the user. As a result, the user can enjoy the data service directly without contacting the LBS server. In fact, the neighbors in this case act as reverse proxy servers, by which the user’s request can be satisfied locally.

(2) If the neighbors do not cache the corresponding service data, then the user has to send a live query to LBS server to obtain the related service. For such a user, we employ *l-diversity* to protect his privacy. More specifically, each user in our solution receives many queries from his peers. Some of the queries can be answered by the user. We denote the set of queries that cannot be answered by the user as *Z*. To protect user’s privacy, we send the user’s real query $Q_{real}$, together with l−1 other queries $Q_{1},Q_{2},\cdots ,Q_{l-1}$ chosen from *Z*, to the LBS server. Here, each query sent to the LBS server by the user has the form of
(1)Q=〈id,loc,q〉
where id represents the identity of the user, loc represents the location of the user, and *q* represents the query interest of the user. Due to our aim being *l-diversity* on user’s location privacy and query privacy, we need to ensure that loc as well as *q* in all *l* queries are different from each other. In other words, for $Q_{real}=\langle id,loc_{real},q_{real}\rangle$ and $Q_{N}=\langle id,loc_{N},q_{N}\rangle \left(N=1,2,\cdots ,l-1\right)$, we have $loc_{i}\neq loc_{j}$, $q_{i}\neq q_{j},\forall i,j\in \{real,1,2,\cdots ,l-1\}$. Note that id in each query *Q* is a pseudonym. For reference convenience, we provide Table 1 to summarize the important notations.

## 4. The Proposed Solution

We begin this section by presenting the system architecture and attacker model. We then describe the proposed Cache-Based Privacy-Preserving (CBPP) solution in detail.

### 4.1. System Architecture and Threat Model

As shown in Figure 1, entities in our system can have two roles, i.e., mobile users and the LBS server. We give a simple explanation for each of them as follows.

Mobile user: A mobile user is someone who holds location-aware (e.g., GPS) mobile devices (e.g., smartphone). In our architecture, each user can position himself and keeps a buffer in his mobile device to record the LBS-related service data. Besides, the same as in [15], each mobile user in our system is equipped with two wireless network interface cards; one of them is dedicated to communicating with the LBS server through the base station, while the other one is devoted to communicating with other peers. A similar multi-interface technique has been used to implement IP multi-homing for stream control transmission protocol (SCTP), in which a machine is installed with multiple network interface cards, and each assigned a different IP address [37]. Similarly, in mobile P2P cooperation environment, mobile users have a network connection to access information from the server, e.g., through a wireless modem or a base station, and the mobile users also have the ability to communicate with other peers via a wireless LAN, e.g., IEEE 802.11 or Bluetooth [38,39,40]. Note that users communicate with each other via single-hop communication and/or via multi-hop routing in a P2P network. Last but not least, each user is assumed to be trusted and concerned about location privacy and query privacy when he seeks service data from the LBS server.

LBS server: An LBS server can be any online location-based service provider. When the LBS server receives queries from users, it searches for the corresponding service data in its database and returns the data to users. Actually, the LBS server computes an answer set that includes the exact answer to the user. Moreover, the LBS server is always seen as untrusted and considered as the adversary in our architecture. Besides, the LBS server can easily get all the background knowledge, and know the privacy protection algorithms used in our system.

### 4.2. Proposed Cbpp Solution

In this section, we present the proposed Cache-Based Privacy-Preserving (CBPP) solution in detail. As we mentioned before, the proposed CBPP solution includes the following two parts. 

#### 4.2.1. Query to Neighbors

When a user *U* needs LBSs, he sends a query with the form of $Q_{U2N}=\langle loc_{U},q_{U},h,t_{U}\rangle$ to his *N* neighbors first to seek for an answer. We denote $loc_{U}$ as the location of *U*, $q_{U}$ as the query interest of *U*, and $t_{U}$ as the time point at which *U* broadcasts $Q_{U2N}$. Algorithm 1 gives the pseudo code of the query to neighbors. Specifically, our query to neighbors algorithm includes the following 4 steps:

(1) *U* first needs to determine $h(=\left\lceil \frac{R_{q}}{R_{h}}\right\rceil )$, which is related to the user’s query range $R_{q}$ and the communication range of a hop $R_{h}$. Actually, a bigger *h* brings higher probability that *U* obtains the required service data. Obviously, the greater *h* is, the more hops *U* can obtain. As a result, *U* has more chances to get the service data from more neighbors. However, a large *h* also leads to more communication costs and service delay. Then, *U* broadcasts the query $Q_{U2N}$ to his *N* neighbors to seek for an answer. *U* listens to the network and waits for the reply from his neighboring peers (Line 3–4 in Algorithm 1).

(2) When *N* receive the query $Q_{U2N}$, *N* first search their buffers, which can be seen as a “Q&A list” in Figure 2, with the keyword $q_{U}$. If *N* have the corresponding answer, then they send a message
(2)$$M=\langle loc_{U},q_{U},answer,t\rangle$$
where answer represents the service data *U* requires, and *t* represents the time when *M* is sent by *N*, to others. Otherwise, *N* discard $Q_{U2N}$. Next, *N* check *h*. If h=1, then *N* stop the algorithm; If h>1, then *N* decrement *h* and broadcast the query $Q_{U2N}$ with the updated h=h−1 to their neighboring peers, who repeat the step (2) (Line 5–19 in Algorithm 1).

(3) For peers who receive *M*, they check whether they are the originator *U* of the query $Q_{U2N}$. Let $loc_{p}$ denote the current position of peer. If $loc_{U}=loc_{p}$ and $q_{U}\in U$ hold simultaneously, then the peer is *U*; Otherwise, the peer continue to broadcast *M* to others (Line 21–28 in Algorithm 1).

(4) When *U* receives *M*, he determines whether *M* has expired. *M* is considered to be expired if the time interval $\left|t-t_{U}\right|$ is beyond Δt, which is a threshold predefined by *U*. If *M* has expired, then *U* discards *M*; Otherwise, *U* gets his required service data (Line 30–35 in Algorithm 1).

**Algorithm 1:** Query to Neighbors Algorithm.
1:**Function** User’s query is answered locally by neighboring peers2:
*// Phase 1: Broadcast user’s query to neighbors*
3:*U* determines a proper $h=\left\lceil \frac{R_{q}}{R_{h}}\right\rceil$;4:*U* broadcasts a query with the form of $Q_{U2N}=\langle loc_{U},q_{U},h,t_{U}\rangle$;5:**for***N* receiving $Q_{U2N}$
**do**6:  **while**
h>1
**do**7:   *N* searches their buffers;8:   **if**
*N* has the corresponding service data **then**9:    Send $M=\langle loc_{U},q_{U},answer,t\rangle$ to neighboring peers;10:   **else**11:    *N* discards $Q_{U2N}$;12:   **end if**13:   h←h−1;14:   *N* broadcasts the query $Q_{U2N}=\langle \;loc_{U},q_{U},h,t_{U}\rangle$;15:  **end while**16:  **if**
h==1
**then**17:   Run lines 7–12;18:  **end if**19:
**end for**
20:
*// Phase 2: Broadcast neighbors’ found data to U*
21:**for** peers receiving *M*
**do**22:  **if**
$loc_{U}=loc_{p}$ & $q_{U}\in U$
**then**23:   Peer is *U*;24:   Break;25:  **else**26:   Continue broadcasting *M* to others;27:  **end if**28:
**end for**
29:
*// Phase 3: Check the validity of M*
30:*U* checks *t* in *M*31:
**if**
$t-t_{U}\leq \Delta t$
**then**
32:  *M* is valid;33:
**else**
34:  *U* discards the message *M*;35:
**end if**



Figure 2 gives a simple running example for the query to neighbors algorithm. In Figure 2, we suppose that there are 10 mobile users who are represented by smartphones, and set the number of hops of neighbors allowed to spread within the P2P network equals to two, i.e., h=2. *U* represents the user who issues a query to neighboring peers, and $N_{i}$ represents the neighbors of *U*. The dotted circle with the same color as the smartphone represents the communication range of this node, and the smartphones with green stars ($N_{1}$, $N_{2}$, $N_{5}$, $N_{7}$, $N_{9}$) represent the neighbors who have the corresponding service data *U* requires.

#### 4.2.2. Query to LBS Server

When a user *U* cannot receive the required service data from his neighboring peers, or the messages *U* received from neighbors are invalid, he has to send a live query to the LBS server. As a result, the LBS server knows some personal information about *U*, such as where he is, what kind of query he submits, etc. However, as we mentioned above, the LBS server is untrusted, and it may threaten $U^{\prime }s$ privacy and security. Therefore, in order to protect the user’s location privacy and query privacy from the LBS server, we propose the query to the LBS server algorithm, the basic idea of which is to utilize *l-diversity*. Algorithm 2 gives the pseudo code of the query to the LBS server, and the technical details are shown as follow.

(1) *U* first needs to decide a proper degree of anonymity *l*, which is closely related to $U^{\prime }s$ location privacy and query privacy. Actually, the greater *l* is, the higher anonymity degree *U* can get. However, a greater *l* can also incur higher communication cost due to the transmission of *l* queries (Line 3 in Algorithm 2).

(2) Generally, *U* receives many queries from his neighboring peers. For these queries, some of them can be answered by *U* utilizing the caching data in his buffer, while the other cannot. *U* then derives a set *Z* of queries that cannot be answered by himself. More specifically, when *U* receives a query $Q_{N2U}$ from his neighbor, *U* first searches his buffer and seeks for the corresponding service data to $Q_{N2U}$. If *U* does not have the corresponding service data, then *U* puts the query $Q_{N2U}$ into *Z* (Line 4–12 in Algorithm 2).

(3) Each query *Q* sent to the LBS server includes $\left(id,loc,q\right)$. To protect users’ sensitive information from the LBS server, we design a query to the LBS server algorithm, which provides *l-diversity* for user’s location information and query interest simultaneously. Specifically, *U* sends the real query $Q_{U}$, together with l−1 other dummy queries $\left(Q_{1},Q_{2},\cdots ,Q_{l-1}\right)$ chosen from *Z*, to the LBS server. We denote the final requiring message sent to the LBS server by *U* as $Q_{U2S}$, then we have
(3)$$Q_{U2S}=\left(Q_{U},Q_{1},Q_{2},\cdots ,Q_{l-1}\right)$$

**Algorithm 2:** Query to LBS Server Algorithm.
1:**Function** User’s query is answered by LBS server2:
*// Phase 1: U derives the set Z of queries not answered by U*
3:*U* determines a proper *l*;4:*U* derives a set Z=⌀;5:The number of elements in *Z* is $z=\left|Z\right|=0$;6:**for** query $Q_{N2U}$ received by *U* from neighbor **do**7:  *U* searches his buffer;8:  **if**
*U* does not have the answer to $Q_{N2U}$
**then**9:   *U* puts $Q_{N2U}$ into *Z*;10:   z←z+1;11:  **end if**12:
**end for**
13:
*// Phase 2: U protects privacy by achieving l−diversity*
14:
**if**
z≥l−1
**then**
15:  *U* constructs set *H* which contains all loc of *Z*, i.e., $H=\left(loc_{1},loc_{2},\cdots ,loc_{z}\right)$;16:  *U* constructs set *I* which contains all *q* of *Z*, i.e., $I=\left(q_{1},q_{2},\cdots ,q_{z}\right)$;17:  *U* constructs set $J=\left(loc_{U}\right)$ and $K=\left(q_{U}\right)$;18:  **for**
i=1; i≤z; i++
**do**19:   **if**
$loc_{i}$∉*J*
**then**20:    Insert $loc_{i}$ into *J*;21:   **end if**22:  **end for**23:  **for**
j=1; j≤z; j++
**do**24:   **if**
$q_{j}$∉*K*
**then**25:    Insert $q_{j}$ into *K*;26:   **end if**27:  **end for**28:  *U* selects a subset $J^{\prime }$ from $J\setminus \{loc_{U}\}$ such that $|J^{\prime }|=l-1$; *$J\setminus \{loc_{U}\}$ is a set whose elements are in *J* but not in $\left\{loc_{U}\right\}$*29:  *U* selects a subset $K^{\prime }$ from $K\setminus \{q_{U}\}$ such that $|K^{\prime }|=l-1$; *$K\setminus \{q_{U}\}$ is a set whose elements are in *K* but not in $\left\{q_{U}\right\}$*30:  **for** each $\left(J^{\prime }\left[i\right],K^{\prime }\left[i\right]\right)\in \left(J^{\prime },K^{\prime }\right)$
**do**31:   $Q_{i}=\left(id_{U},J^{\prime }\left[i\right],K^{\prime }\left[i\right]\right)$;32:  **end for**33:  *U* replaces $id_{U}\in Q_{U}$ and $id_{i}\in Q_{i}$ with a pseudonym id;34:  *U* sets $Q_{U2S}=\left(Q_{U},Q_{1},Q_{2},\cdots ,Q_{l-1}\right)$;35:
**else**
36:  *U* waits for more queries and repeats line 15–34;37:
**end if**
38:*U* sends required message $Q_{U2S}$ to LBS server;


To effectively achieve *l-diversity* on $U^{\prime }s$ location information and query interest, dummy queries $Q_{i=1,2,\cdots ,l-1}$ need to be chosen based on the given regulation as follows.

When *U* selects a dummy query $Q_{i}$ in *Z*, due to our aim being *l-diversity* on user’s location privacy and query privacy, we need to ensure that loc as well as *q* in all *l* queries are different from each other. In other words, for $Q_{U}=\left(id_{U},loc_{U},q_{U}\right)$ and $Q_{i}=\langle id_{i},loc_{i},q_{i}\rangle \left(i=1,2,\cdots ,l-1\right)$, we have $loc_{i}\neq loc_{j}$, $q_{i}\neq q_{j},\forall i,j\in \left\{U,1,2,\cdots ,l-1\right\}$ (Line 14–32 in Algorithm 2).

Last but not least, *U* replaces his real identity $id_{U}$, as well as all dummy identities $id_{i}$ in $Q_{i}$, with a pseudonym id. Finally, *U* sends the required message
(4)$$Q_{U2S}=\left(Q_{U},Q_{1},Q_{2},\cdots ,Q_{l-1}\right)$$
to LBS server (Line 33–38 in Algorithm 2).

We illustrate our query to the LBS server algorithm based on the example shown in Figure 3. In Figure 3, there are 5 mobile users represented by smartphones. *U* represents the user who issues a query to the LBS server, and $N_{i=1,2,3,4}$ represent the neighbors of *U*. The dotted circle represents the communication range of *U*. As shown in Figure 3, suppose l=4, which means that *U* needs to satisfy 4-*diversity* for his location information and query interest. Thus, *U* selects the other 3 dummy queries $Q_{1},Q_{2},Q_{4}$ received from $N_{1},N_{2},N_{4}$ respectively according to proposed query to the LBS server algorithm. Finally, the required message sent to the LBS server by *U* is $Q_{U2S}=\left(Q_{U},Q_{1},Q_{2},Q_{4}\right)$. When the LBS server receives $Q_{U2S}$, it searches its database and transmits the answers set, which includes the answer $D_{U}$ to $Q_{U}$ as well as the answer $D_{i=1,2,4}$ to $Q_{i=1,2,4}$, back to *U*.

## 5. Solution Analysis

In this section, we first provide the security analysis to proposed CBPP solution. Then, we analyze the impact of the cache hit ratio of the proposed CBPP solution.

### 5.1. Security Analysis

The aim of the adversary in the LBS environment is to obtain a user’s sensitive information, such as his location information and query interest. In our work, we assume that the adversary has compromised and controls the LBS server, thus it knows all the information of the user. We also assume that the adversary knows the algorithm we use. Our analysis focus on how the proposed CBPP solution can prevent the possible privacy leakage from the LBS server and other users in the system.

*Case 1: The LBS server does not know the location information and the query interest of the user*.

On the one hand, a user sends a query to his neighbors first to seek for an answer when he needs LBSs. If the neighbors have the corresponding service data, then they send the answer back to the user. In this way, the user can enjoy the LBSs without contacting the LBS server. Thus, the LBS server cannot obtain any information about the user. On the other hand, if the user cannot get the corresponding data service from his neighbors, then he needs to send a live query to the LBS server. In this case, we employ the *l-diversity* on both the location information and the query interest of the user. Due to the *l-diversity* principle, the LBS server cannot deduce the exact location information and the exact query interest of the user with a probability larger than $\frac{1}{l}$. As a result, the LBS server does not know the location information and the query interest of the user.

*Case 2: The user does not disclose other users’ location information and query interest*.

In our work, when a user wants LBSs, he first sends the query message to his neighbors to seek for the answer. In this way, neighbors can obtain some sensitive information of the user, such as his location information and query interest. However, as we just mentioned above, each node in the P2P network of our work is assumed to be trusted. Therefore, each user cannot disclose the sensitive information to others. Actually, when neighbors receive the query message from the user, they just search the buffer with the keyword. If they have the corresponding answer, they then correctly return the answer to the user. Otherwise, they continue to transmit the message to others.

### 5.2. Cache Hit Ratio Analysis

In our work, a cache hit occurs when the requested service data of a user can be found in his neighbors’ caches. In other words, if a user can get the service data from his neighbors, then a cache hit occurs. Otherwise, if a user cannot enjoy the LBSs locally and needs to send a live query to the LBS server to get the service data, then a cache miss occurs. To protect the user’s privacy against the LBS server, we need to improve the cache hit ratio. A high cache hit ratio means that a user can get the requested service data from his neighbors directly with a higher probability. By improving the cache hit ratio, the user reduces the number of queries sent to the LBS server, which makes the LBS server know less information about the user. We analyze the cache hit ratio of proposed CBPP solution from the following two aspects.

*Case 1: Query to neighbors algorithm can improve the cache hit ratio*.

When a user needs LBSs, he first sends a query message to his neighbors. If neighbors store the corresponding service data in their buffers, then they correctly broadcast the data to the neighboring peers. In this way, a large number of peers (including the user) in the P2P network can obtain the service data. We argue that, the more data a peer in the P2P network stores in his buffer, the higher cache hit ratio other peers can get, because other peers have greater chance to get the requested service data from this peer. In the extreme situation, if a peer has all the global data, then he is equivalent to the LBS server. Thus, the query to neighbors algorithm in our solution can effectively improve the cache hit ratio.

*Case 2: Query to LBS server algorithm can improve the cache hit ratio*.

When a user cannot get the requested service data from his neighbors, he must contact the LBS server to obtain the data. To protect the user’s privacy, in our solution, the user sends a set of query $Q_{U2S}=\left(Q_{U},Q_{1},Q_{2},\cdots ,Q_{l-1}\right)$, which includes $U^{\prime }s$ real query $Q_{U}$ and l−1 dummy queries $Q_{i=1,2,\cdots ,l-1}$, to the LBS server. Actually, the $Q_{i=1,2,\cdots ,l-1}$ are the queries that come from the other peers and cannot be answered by the user. In other words, when a user needs to contact the LBS server, he selects l−1 dummy queries from all his received queries to achieve the *l-diversity*. More importantly, the user cannot find the corresponding service data in his buffer for the l−1 dummy queries. When the user receives the answers set from the LBS server, he not only gets his requested answer, but also the answers of the l−1 dummy queries. In this way, the user can answer the same queries as the l−1 dummy queries in the future, which means that the query to the LBS server algorithm in our solution can improve the cache hit ratio.

## 6. Performance Evaluation

### 6.1. Simulation Setup

To evaluate the performance of CBPP, we implement its algorithms in a Windows 10 desktop computer. Each result in our experiments is an average of 100 iterations to make them more exact. We deploy 1000 mobile users in the P2P network. We also choose 10 users to seek the same service information every 1 minute, the query radius is 1km and the system is run for 2 h. Moreover, in our following experiments, *l* is related to the *l-diversity* and is set by the user, *t* is the evaluation time, *h* is the maximal number of hops in the P2P network. We compare CBPP with three existing schemes: *DLS* [20], *Mobicache* [21], and *MobiCrowd* [22]. Note that *DLS* does not use the caching but the other two schemes do.

### 6.2. Evaluation Results

Table 2 presents a simple comparison of our solution with several previous works on some privacy properties. Clearly, most existing works fail to protect users’ location and query privacy simultaneously, and most of existing works do not use the caching. In the following, we present our evaluation results in detail.

*(1) Effects of l on privacy*. We use the *l-diversity* to measure the location privacy for users who send live queries to the LBS server. Actually, the principle of *l-diversity* and *k-anonymity* is the same, they both utilize some dummy locations to confuse the LBS server. Moreover, we let h=5 and t=15 min. As shown in Figure 4a, the privacy degree of *DLS* [20] is the worst since it does not use the caching. *MobiCrowd* [22] performs better since it caches the obtained service data to serve others. However, *MobiCrowd* always submits users’ real locations to the LBS server, and it fails to protect location privacy of the users who have to send live queries. Thus, we can see from Figure 4b that the real locations of users are always exposed and the probability of exposing the real location is 1. We can also find in Figure 4a that our proposed CBPP solution has the highest privacy degree in the four schemes. Although the probability of exposing real location in proposed CBPP solution, *DLS* and *Mobicache* [21] are very close in Figure 4b, we protect the location privacy and the query privacy of users simultaneously.

*(2) Effects of l on cache*. Figure 5 illustrates how *l* affects the cache hit ratio when h=5, t=15 min. *DLS* [20] does not consider caching, thus its cache hit ratio is equal to 0. Since *MobiCrowd* [22] does not consider dummy locations, thus its cache hit ratio stays at a fixed level (around 30%). *Mobicache* [21] uses dummy locations, which result in a higher cache hit ratio. This is because dummies can make more contributions to improve the chance that the other users get the requested service data. In our solution, we carefully select dummy locations and achieve the highest cache hit ratio.

*(3) Effects of t on cache*. We present how cache hit ratio changes with the simulation time *t* in Figure 6. In this experiment, we let the number of hops h=5, and let l=10. We can find in Figure 6 that the cache hit ratio of *Mobicache* [21], *MobiCrowd* [22] and proposed CBPP solution increases with *t* since more service data is cached as time goes by. However, the cache hit ratio of *DLS* [20] is 0. This is because the *DLS* scheme does not utilize the caching. We can also see that in all schemes, proposed CBPP solution has the highest cache hit ratio.

*(4) Effects of t on privacy*. We evaluate the impact of the simulation time *t* on privacy degree, which is shown in Figure 7. We also let the number of hops h=5, and let l=10. Generally, the privacy degree in *Mobicache* [21], *MobiCrowd* [22] and proposed CBPP solution increases with the simulation time *t*. The *DLS* [20] stays an almost fixed value (around 3). Actually, as time goes on, more data is cached and fewer queries are sent to the LBS server, which greatly enhances user privacy. Proposed CBPP solution outperforms all other schemes since it can cache more service data as time goes by.

*(5) Effects of h on cache*. Figure 8 shows the relationship between the cache hit ratio and the number of hops *h*. In this case, we let l=10 and t=15 minutes, and we evaluate the number of hops form 1 to 5. Obviously, in most cases, the more hops a user can obtain, the higher cache hit ratio the user can get. The reason is that, when increasing *h*, a user has more opportunities to get his requested service data from his neighboring peers in the P2P network. We can also see that the cache hit ratio of *DLS* [20] has nothing to do with the number of hops *h*, and the cache hit ratio of *DLS* is 0 because of its ignorance of caching.

## 7. Conclusions and Future Work

### 7.1. Conclusions

In this paper, we proposed a Cache-Based Privacy-Preserving (CBPP) solution for users in LBSs. Different from the previous approaches, CBPP considers location privacy and query privacy simultaneously, and also avoids the problem of a TTP server by having users collaborating with each other to improve the privacy. The key idea of CBPP is that each user caches the service data obtained from the LBS server or the neighbors in his mobile device, and uses the cached data to answer the later queries issued by others. Specifically, when a user wants LBSs, he needs to broadcast a query to his neighbors first to seek for an answer. The user only contacts the LBS server if he cannot get the required service information from the neighbors. Actually, each user now can be seen roughly as a micro TTP server. In this way, a user reduces the number of queries sent to the LBS server, and remains hidden from the LBS server. To users who have to send live queries to the LBS server, we employ *l-diversity* to further protect their privacy. Evaluation results show that CBPP can effectively protect users’ location privacy and query privacy with a higher cache hit ratio and better quality of service.

### 7.2. Future Work

This section discusses some potential challenges and future directions of our work. As mentioned before in this paper, users are assumed to be trusted and directly share location information with each other. We argue that, however, this assumption is unrealistic since users could not be fully trusted in practice. Typically, a malicious user may pretend to be a normal one and collect the locations of the neighboring peers, which leads to the malicious user tracking peers directly or releasing their personal information to third parties. Actually, we argue that the exposure of user’s exact location to any entity would reveal his personal sensitive information. In this sense, the proposed solution has limited applications, and location cloaking without exposing the accurate user location to his neighboring peers is urgently needed.

In order to solve this problem, we propose a sketchy solution, which is used against the untrusted neighboring peers in mobile peer-to-peer (P2P) environment. In our sketchy solution, no trust relationship is assumed among users. The main idea of it is that a user hides his exact location in the request within a cloaking region (CR), which is transformed by the user with the Hilbert curve, and broadcasts the request to his neighbors to seek for an answer. In this way, neighbors just realize that the user is in the CR, but cannot pinpoint his accurate location, which greatly improves the user’s location privacy. After receiving the request from the user, neighbors utilize the Voronoi Diagram (VD) to find the corresponding service data based on the query interest, and then send the service data back to the user. The user only contacts the LBS server when he cannot obtain the required service data from his neighbors.

Actually, we are now in the middle of this work, i.e., we are now utilizing the Hilbert curve and Voronoi Diagram to protect user’s location privacy and execute the KNN query on an area. Security analysis and evaluation results show that not only can the proposed sketchy solution effectively protect users’ location privacy, but it can also provide better quality of service (QoS) for KNN query without the accurate location information of the user.

## Figures and Tables

**Figure 1 sensors-20-04651-f001:**
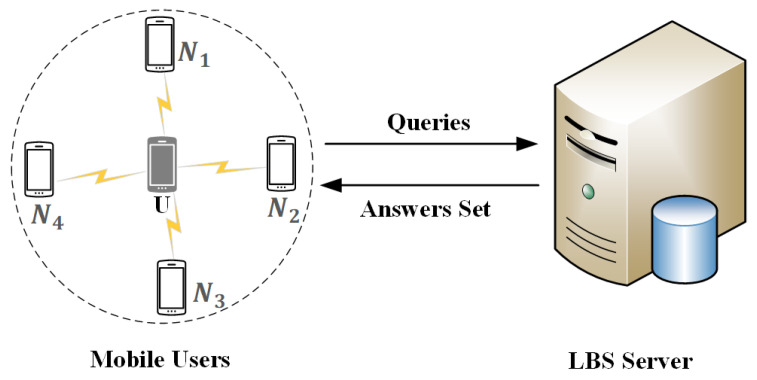
The system architecture.

**Figure 2 sensors-20-04651-f002:**
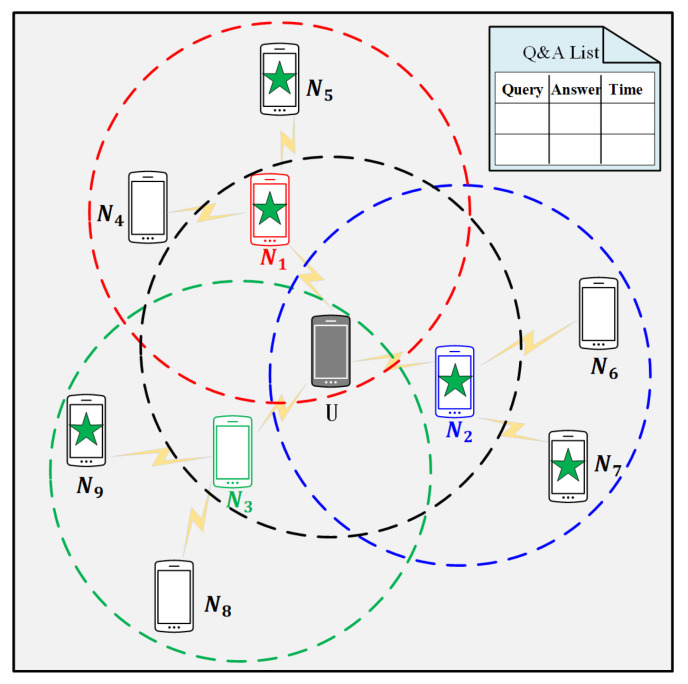
Query to neighboring peers.

**Figure 3 sensors-20-04651-f003:**
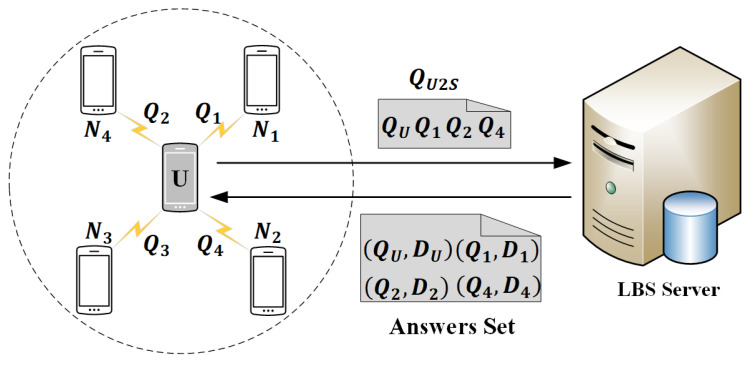
Query to the LBS server.

**Figure 4 sensors-20-04651-f004:**
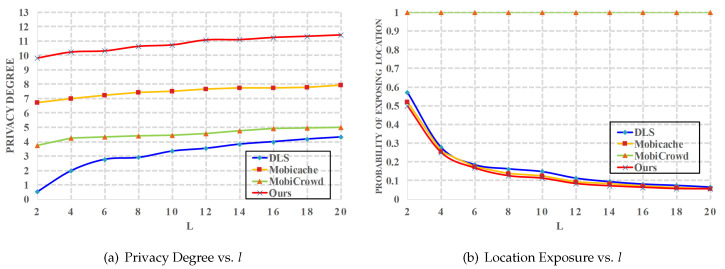
Privacy vs. *l*.

**Figure 5 sensors-20-04651-f005:**
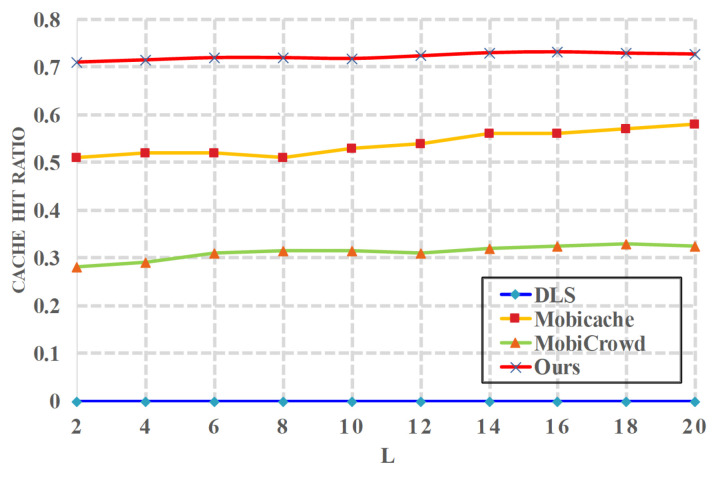
Cache Hit Ratio vs. *l*.

**Figure 6 sensors-20-04651-f006:**
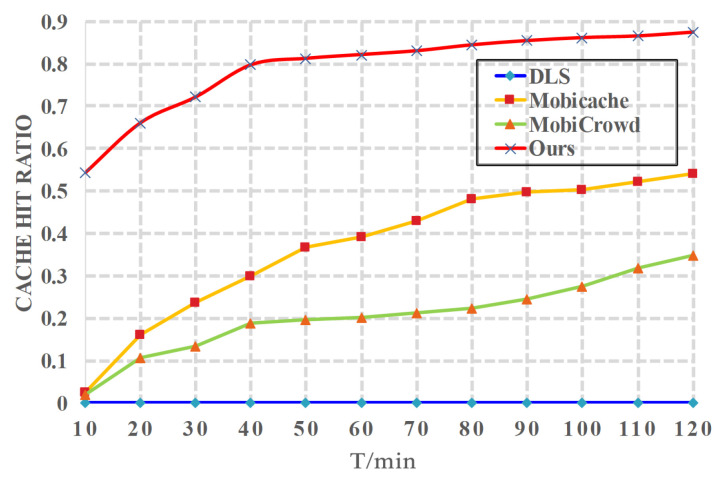
Cache Hit Ratio vs. *t*.

**Figure 7 sensors-20-04651-f007:**
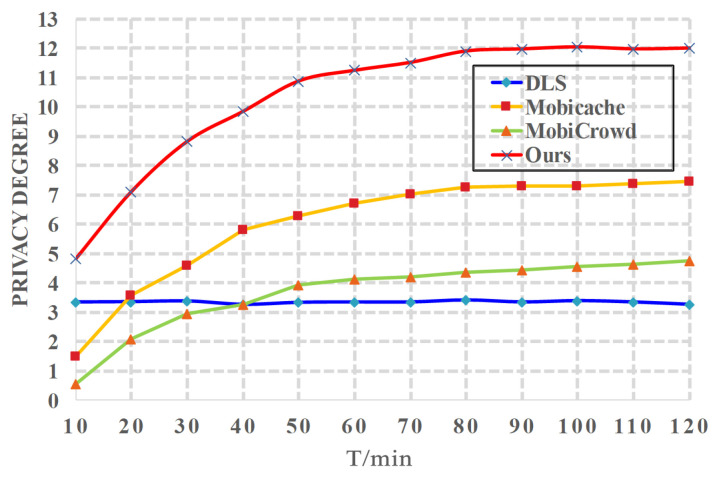
Privacy Degree vs. *t*.

**Figure 8 sensors-20-04651-f008:**
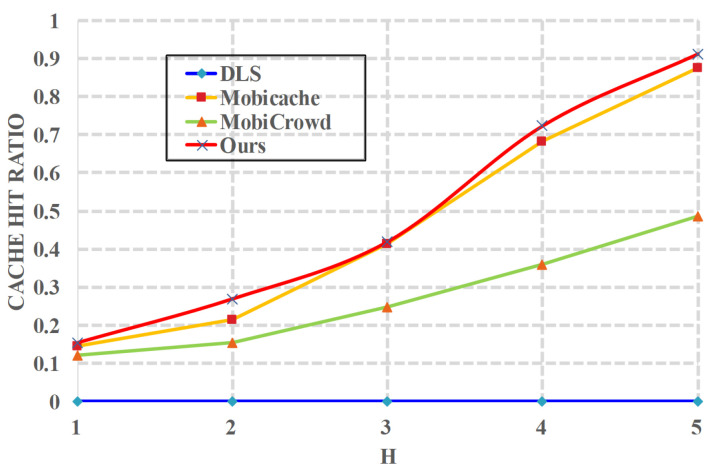
Cache Hit Ratio vs. *h*.

**Table 1 sensors-20-04651-t001:** Notation reference table.

Notation	Meaning
*U*	User who submits query to neighbors or LBS server
id	Identity (pseudonym) of the node in P2P network
loc	Location of the node
*q*	Query interest of the node
*h*	Number of hops
$Q_{U2N}$	Query sent to Neighbor by the user
$Q_{U2S}$	Query sent to LBS server by the user
*l*	Anonymity level
*Z*	Set of queries not answered by *U*
*N*	Neighbor of the user
*M*	Message sent to *U* by his neighboring peers

**Table 2 sensors-20-04651-t002:** Comparison of privacy properties.

	DLS [20]	TTcloak [2]	PPCP [26]	MobiCrowd [22]	MobiCache [21]	CBPP
TTP server	×	×	✓	×	×	×
Location privacy	✓	✓	✓	✓	✓	✓
Query privacy	×	✓	✓	×	×	✓
Use cache	×	×	×	✓	✓	✓
Improve cache hit ratio	×	×	×	×	✓	✓

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
