# Peer review of "Cache-Based Privacy Preserving Solution for Location and Content Protection in Location-Based Services"

_sensors, 2020, doi:10.3390/s20164651_

Round 1

Reviewer 1 Report

This submissions addresses the problem of user privacy in centralised
location-based services. The authors focus on two aspects of location-based
services, where sensitive user data is stored or processed on untrusted
servers, namely the actual location information but also meta-date
throughout user queries. The authors propose a cache-based solution which
relies on querying neighbouring devices before contacting the server. The
solution is evaluated based on simulated scenarios and through a security
argument, where the authors show that location privacy and query privacy
are protected while a sufficiently hight cache hit ratio improves query
efficiency.

The main shortcoming of the article is that the approach relies on mobile
peer-to-peer communication between neighbouring devices, the setup of which
is never explained in the paper. My understanding -- and I am no expert in
this -- is that some trusted entity with knowledge of (at least) routing
information and (probably also) the location of neighbouring devices in a
1km-range (simulation scenario, p. 11) would be required to establish this
peer-to-peer communication. Thus, the author's claim to be able to work
without a Trusted Third Party appears to be false. I could imagine
peer-to-peer communication to be established via BlueTooth, which would
however, not provide for the communication range used in the simulation
scenario.

The authors correctly identify malicious peers/neighbours as a serious
threat to their solution, albeit without proposing even a sketchy solution
to this. Could such a solution be outlined? I was thinking of Trusted
Execution on mobile devices to provide attestation for a basic peer-to-peer
endpoint for location-based services?

Overall the paper is well written but not self-contained. Specifically, an
explanation of the technology and privacy assumptions for the peer-to-peer
infrastructure are missing, which makes it difficult to assess the technical
soundness of the paper. I recommend to request a re-submission from the
authors.

Reviewer 2 Report

The submitted manuscript, titled “Cache-Based Privacy Preserving Solution for Location and Content Protection in Location-Based Services”, is focused on the Location-Based Services (LBSs), software services that exploit geographic data and information in order to provide a series of services/information to users.
In such a context, the authors underline the issues related to the privacy, since these services involve personal information, which are usually stored using untrusted servers.
In order to face this problem, they propose a Cache-Based Privacy-Preserving (CBPP) approach for the LBSs, which should be able to protects both location privacy and query privacy, avoiding the problem related to the Trusted Third-Party (TTP) server (i.e., many state-of-the-art solutions rely heavily on this kind of server) by having users collaborating with each other in a mobile peer-to-peer (P2P) environment.
The authors claim that the proposed CBPP approach effectively protect users’ location and query privacy, presenting a lower communication cost and better quality of service.

Although the manuscript is well written both in terms of technical content and organization, I suggest to the author a careful re-reading of it in order to fix some minor typos and/or grammar forms, such as, for instance: "anonymizing techniques" instead of "anonymization techniques"; “For such user” instead of “For such a user”; “while” instead of “whereas” in several occasions; and so on.

In the "Introduction" and "Related Work” sections the author should cite and discuss further papers focused on information very close or directly related to the research field taken into account.
In addition, in these sections the authors should avoid mixing literature information with information related to the proposed solution, using for this a specific section/subsection.

The references do not appear very updated (only 6 out of 20 are after 2015), so the author should check if there are more recent works among those he mentioned and, in any case, accordingly to my previous observation, they should add and discuss additional works very close or directly related to the domain taken into account, as follows:

(-) Qiu, Y., Liu, Y., Li, X., & Chen, J. (2020). A Novel Location Privacy-Preserving Approach Based on Blockchain. Sensors, 20(12), 3519.
(-) Konings, D., Parr, B., Alam, F., & Lai, E. M. K. (2018). Falcon: Fused application of light based positioning coupled with onboard network localization. Ieee Access, 6, 36155-36167.
(-) Amoretti, M., Brambilla, G., Medioli, F., & Zanichelli, F. (2018, July). Blockchain-based proof of location. In 2018 IEEE International Conference on Software Quality, Reliability and Security Companion (QRS-C) (pp. 146-153). IEEE.
(-) Saia, R., Carta, S., Recupero, D. R., & Fenu, G. (2019, February). Internet of Entities (IoE): A Blockchain-based Distributed Paradigm for Data Exchange between Wireless-based Devices. In SENSORNETS (pp. 77-84).
(-) Zhang, S., Mao, X., Choo, K. K. R., Peng, T., & Wang, G. (2020). A trajectory privacy-preserving scheme based on a dual-K mechanism for continuous location-based services. Information Sciences, 527, 406-419.
(-) Habibzadeh, H., Nussbaum, B. H., Anjomshoa, F., Kantarci, B., & Soyata, T. (2019). A survey on cybersecurity, data privacy, and policy issues in cyber-physical system deployments in smart cities. Sustainable Cities and Society, 50, 101660.

The formal approach followed by the authors in order to describe the proposed solution has been presented in a quite clear form to the readers, but the algorithms 1 and 2 appear a bit confused due to the number of code lines and the not-clear notation adopted in some of them (e.g., lines 28 and 29 of the Algorithm 2).
The pseudo-code form should also be improved by the authors, adopting the pseudo-code formalism largely used in literature.
In order to improve the readability of the manuscript, I also suggest moving some formulas from the inline-form to the equation-form.

The authors should change the order of the “Conclusion” and “Future Work” sections or (better) they should merge these sections into a single “Conclusion and Future Work” sections, since it is illogical to discuss the future work before their conclusions.
Their conclusion well summarizes the idea behind the proposed solution, but the authors should better underline the related advantages with regard to the state-of-the-art solutions.
